# What Can Trigger Spontaneous Regression of Breast Cancer?

**DOI:** 10.3390/diagnostics13071224

**Published:** 2023-03-24

**Authors:** Nicoletta D’Alessandris, Angela Santoro, Damiano Arciuolo, Giuseppe Angelico, Michele Valente, Giulia Scaglione, Stefania Sfregola, Angela Carlino, Elena Navarra, Antonino Mulè, Gian Franco Zannoni

**Affiliations:** 1Pathology Unit, Department of Woman and Child’s Health and Public Health Sciences, Fondazione Policlinico Universitario Agostino Gemelli IRCCS, 00168 Rome, Italy; 2Department of Medical and Surgical Sciences and Advanced Technologies G. F. Ingrassia, Anatomic Pathology, University of Catania, 95123 Catania, Italy; 3Pathology Institute, Catholic University of Sacred Heart, 00168 Rome, Italy

**Keywords:** breast cancer, spontaneous regression, pathology, chemotherapy, immune response

## Abstract

Background: Spontaneous regression of tumors is a rare phenomenon in which cancer volume is reduced or, alternatively, a tumor completely disappears in the absence of any pharmacological treatment. This phenomenon has previously been described in several tumors, such as neuroblastomas, testicular malignancies, renal cell carcinomas, melanomas, and lymphomas. Spontaneous remission has also been documented in breast cancer; however, it represents an extremely rare and poorly understood phenomenon, with only a few reported cases in the literature. Methods: We herein report two cases of breast cancer that showed spontaneous tumor regression in the surgical specimen after a pathologically confirmed diagnosis of invasive breast cancer in core needle biopsy samples. Results: Macroscopically, both the surgical samples revealed a whitish, fibrous area with a rubbery consistency. On histological examination, diffuse fibrous tissue, hemosiderin deposition, and chronic inflammation were observed. The first case showed the complete disappearance of the tumor, whereas the second case showed just a small (3 mm), residual nest of neoplastic cells. Conclusions: Although spontaneous regression of breast cancer is a rare event, it is important to know that it might happen. It is also of great importance to try to better explain, over time, its underlying mechanism. This knowledge could help us to further develop cancer prevention methods and predict the clinical course of these kinds of neoplasms.

## 1. Introduction

Spontaneous regression (SR) of tumors is a rare event that is defined by F.W. Stewart as “the partial or complete disappearance of a proven malignant tumor in the absence of all treatment, or in the presence of therapy which is considered inadequate to exert a significant influence on neoplastic disease” [1,2,3]. Despite being an extremely rare phenomenon, occurring in fewer than 1 in 100,000 cases of cancer, there have been several documented cases of SR in the literature [1,2,3,4,5,6,7,8,9,10,11,12]. The most common tumors that have been reported to undergo SR are neuroblastomas, testicular malignancies, renal cell carcinomas, melanomas, and lymphomas [2,4,5,6,7,8,9,10,11,12]. Rare cases of SR have also been reported in lung and breast cancer [2,4,5,6,7,8,9,10,11,12]. To date, several theories about the mechanisms involved in SR have been postulated; however, its possible causes and clinical significance are not fully understood.

Immune system activation, biopsy procedures, and changes in the tumor microenvironment have been shown to play a crucial role in both tumor growth and in tumor regression [2,5,6,7,8,9,10,11,12].

Breast cancer (BC) remission was reported for the first time by Muir in 1934 [3]. We herein report two additional cases of breast cancer that showed spontaneous tumor regression in the surgical specimen after a pathologically confirmed diagnosis of invasive breast cancer in core needle biopsy samples. The first case showed complete tumor regression, while the second case showed just a small (3 mm), residual nest of neoplastic cells. The present report provides further evidence of the possibility of SR in breast cancer; moreover, a discussion of the possible mechanisms that trigger the self-healing of the tumors, with a particular focus on the tumor microenvironment and immune response, are also provided.

## 2. Results

### 2.1. Cases Presentation

#### 2.1.1. First Case

A 76-year-old woman was referred to our hospital for a palpable lump in her left breast. Mammographic examination demonstrated a well-defined mass, 3 cm in diameter at its longest, without lymph node involvement. Cytological examination of the fine-needle aspiration revealed numerous atypical epithelial clusters with hyperchromatic nuclei, and a diagnosis of high-grade BC was made (Figure 1). No phlogosis was observed. She underwent lumpectomy plus sentinel node biopsy.

Gross examination of the surgical specimen revealed a 3 cm, whitish area with a rubbery consistency located at the previous bioptic site. Histologically, after a careful sampling of the entire specimen, diffuse fibrosis without residual neoplastic cells was observed (Figure 1). Moreover, hemosiderin deposition and diffuse chronic inflammation were observed. Neither lymphovascular space invasion nor foci of intraductal carcinoma was observed. Immunohistochemistry was performed to further characterize the inflammatory cells observed within the fibrous tissue. The majority of inflammatory cells were represented by T lymphocytes; however, no relevant differences were observed between T helper and T cytotoxic lymphocytes since diffuse positivity was observed for both CD4 and CD8 (Figure 2). Several macrophages highlighted by CD68 were also encountered. On the other hand, both neoplastic and inflammatory cells failed to show PD-L1 expression (Figure 2).

#### 2.1.2. Second Case

An 80-year-old woman was referred to our hospital after a screening ultrasonography revealed an isoechoic mass measuring 4 cm in diameter at its longest on her left breast. Core needle biopsy was performed. Pathological examination revealed triple-negative BC with medullary-like features and multifocal inflammation. She underwent radical mastectomy plus axillary lymphadenectomy (Figure 1).

Gross examination of the surgical specimen revealed a 4 cm, whitish area with a rubbery consistency located at the previous bioptic site. Histologically, after a careful sampling of the entire specimen, diffuse fibro-inflammatory changes associated with a single residual focus of neoplastic cells measuring 3 mm in size were observed (Figure 1). Axillary lymph nodes appeared uninvolved; moreover, foci of intraductal carcinoma and lymphovascular spread of neoplastic cells were not observed.

Immunohistochemical studies in the inflammatory component revealed similar result to the previous case: the majority of inflammatory cells were T lymphocytes with a similar distribution between T helper and T cytotoxic lymphocytes (Figure 2). On the other hand, CD68-positive macrophages were not observed. Interestingly, the residual neoplastic cell component demonstrated membrane PD-L1 staining; moreover, membrane and/or cytoplasmic staining for PD-L1 was observed in the inflammatory (lymphocytes and macrophages) component (Figure 2).

### 2.2. Literature Search of Breast Cancer–Spontaneous Regression

A thorough search in PubMed was performed in order to find all cases of epithelial BC–SR published so far (December 2022). Twenty-nine reported cases were found [13,14,15,16,17,18,19,20,21]. The results of our literature search are summarized in Table 1.

#### Inclusion/Exclusion Criteria

Inclusion criteria for studies selection were (i) reporting SR of a histologically confirmed malignant epithelial tumor of the breast, (ii) no history of adjuvant/neoadjuvant chemotherapy and (iii) no history of hormonal treatment.

Exclusion criteria for studies selection were (i) reporting SR of primary tumors outside the breast and (ii) reporting regression of breast metastases.

## 3. Discussion

Spontaneous regression of cancer is a rare, well documented, and still surprising phenomenon. It can occur in primary tumors or involve their metastases and has been reported in several anatomical locations, including the breast [1,2,3]. While SR can occur throughout the body, it is most commonly reported in renal cell carcinoma, melanoma, and neuroblastoma [2,4,5,6,7,8,9,10,11,12]. The frequency of SR varies based on the type of cancer, occurring at a rate of about 1 out of every 60,000 to 100,000 cases of all cancers [2,4,5,6,7,8,9,10,11,12]. The mechanisms of SR are not fully understood, but several putative mechanisms have been proposed, including inflammation, apoptosis, ischemia, and immunological responses [2,4,5,6,7,8,9,10,11,12,13]. Moreover, additional mechanisms suggested to explain the phenomenon of SR include epigenetic modifications, hormonal responses, oncogenes, tumor suppressors, cytokines, growth factors, and psychological factors [2,4,5,6,7,8,9,10,11,12,13].

### 3.1. Fever and/or Acute Infection

Like normal tissues, tumors may be involved in various infections resulting in cell death. Infection, as a potential trigger of tumor regression, was considered many centuries ago in the Ebers Papyrus, the greatest Egyptian medical document. It was suggested that applying a poultice to the tumoral area, followed by an incision, induced an infection that would promote tumor regression [22]. In the second half of 1200s, an Italian priest, Peregrine Laziosi, was suffering from a tumor on his tibia, but the development of a cutaneous infection determined the complete disappearance of his neoplasm [23]. Since that case, several other tumor regression phenomena following infections have been reported. Kutzner, in 1889, described the case of an 18-year-old girl with a round cell sarcoma of the breast with many metastatic deposits in the subcutaneous tissue; all the deposits disappeared after an acute lung infection [6]. Other authors reported the case of a 59-year-old woman whose recurrent breast cancer temporarily receded when tuberculosis developed [6].

Studies have demonstrated that viral infections can trigger interferon production and exert immunomodulatory effects, including the activation of IL-2 receptors in some cancers. The recently observed regression of tumors in response to COVID-19 vaccination and infection highlights the important role of immunogenicity in the involution of GI cancers [11,24].

### 3.2. Incomplete Surgical Removal

Partial removal of tumor mass can be followed by spontaneous regression when the remaining unexcised tissue contains only inflammatory cells that can lead to an erroneous diagnosis of a residual tumor. Some authors also considered the role of the anesthetic used during surgery that can affect lipid metabolism and slow tumor growth [6]. Moreover, partial excision of a tumor impairs the blood supply to the remaining malignant tissue. This is the reason why many SRs have been attributed to biopsy procedures [2,6,10].

### 3.3. Endocrine Influences

Hormonal alterations (menopausal or pregnancy effects, dietary changes, and cessation of oral contraceptives) may have a beneficial effect on tumoral mass [6]. In this regard, studies have shown that pregnancy may contribute to tumor regression in some cases of neuroendocrine tumors [6,25].

### 3.4. Unusual Sensitivity to Inadequate Therapy

SR has been observed even when an unusual response of the neoplasm to ineffective treatment is noted. In this regard, radiation therapy reduces the supply of blood and nutrients to the tumor, so necrosis and/or vascular insufficiency could slow tumor growth. In these cases, despite the fact that treatment is inadequate to cure the disease, the tumor is considered spontaneously regressed [2,6,13].

### 3.5. Immunity

The potential role of the immune response in tumor regression has important therapeutic implications [2,5,6,10,11]. Indeed, SR cases associated with infections have led to the development of various anti-cancer therapies that aim to enhance the immune system’s ability to identify and eliminate cancer cells [2,5,6]. The preferred hypothesis among all those proposed is that cancer cells induce a spontaneous T-cell-mediated response. Ohara et al. defined the concept of “Immunogenic cell death” (ICD) mediated by dendritic cells and T lymphocytes [2,5,6,10,11]. Moreover, immune cell infiltrations and signaling cascades are postulated to cause tumor cell death and necrosis via the release of cytokines and interferons, particularly IL-2, IL-6, and IL-8 [2,6,10,11].

### 3.6. Ischemia

Ischemic models of regression have been proposed to cause the death of tumor cells by limiting blood supply and perfusion [10,26,27,28]. Hypoperfusion, rapid and unpredictable growth, anti-angiogenic factors, and vascular alterations are all possible factors that can lead to SR [10,26,27,28]. Moreover, diagnostic biopsy procedures, tumor ablations, and angiographic techniques have been specifically attributed to SR as they impede the delivery of essential nutrients and oxygenation to malignant tissue while acting as a therapeutic vaccine [10,26,27,28].

### 3.7. Spontaneous Regression in Breast Cancer

Although breast carcinoma is a common malignancy, cases of SR described in the literature are extremely rare [3,6,13,14,15,16,17,18,19,20,21]. Hundreds of tumor regressions published so far have been reviewed; however, to date, only twenty-nine cases of regressed epithelial breast cancer can be considered reliable [3,6,13,14,15,16,17,18,19,20,21]. We noticed that most of the patients were female in a wide age range. It was not possible to trace the precise histological diagnoses for all cases, but we know that at least four of those documented were triple-negative breast cancer (TNBC). Various authors have hypothesized that tumor regression may have been triggered by disparate mechanisms. In particular, both Cserni and Ohara assumed the role of metformin, an anti-diabetic, in SR [13,19]. The exact mechanism behind this “miraculous” phenomenon still remains poorly understood. Given the current interest in anti-tumor immunity and its stimulation with immunotherapy and immune checkpoint inhibitors, this event deserves special attention. The preferred hypothesis is that tumor disappearance is the result of spontaneous T-cell-mediated response [29,30,31]. However, all the above-mentioned factors (infection, trauma, surgery, etc.) could be potential triggers for anti-tumor immune reaction, acting alone or together. The role of infection was the first to be postulated [3,5,6,10,11]. Dussan et al. reported a case of BC that regressed after an arm injury [16].

### 3.8. Proposed Mechanisms of Spontaneous Regression in Our Cases

Based on the clinicopathological features of the cases herein presented, we can hypothesize two different biological mechanisms.

The first case showed fibro-inflammatory changes without evidence of residual neoplastic cells. In our opinion, local trauma induced by fine-needle aspiration may have triggered the immune response following the possible release of a segregated tumor antigen. Moreover, following the initial diagnosis of breast cancer, the first patient that was referred to us started a dietary regimen based on caloric restriction and on the use of Artemisia annua herbal preparations; however, their real effectiveness remains uncertain.

On the other hand, the second case showed PD-L1 expression in the inflammatory and residual neoplastic components. Programmed death ligand (PDL1) is a co-inhibitory receptor located on CD8+ T cells that can suppress the immune response against tumor cells by binding to PD-L1 [32]. Since its expression is generally induced in response to the host’s anti-tumor immune response, we can postulate that an unknown traumatic event, probably related to the biopsy procedure, may have triggered an excessive immune response leading to tumor regression.

## 4. Conclusions

Spontaneous cancer healing remains a challenging issue. Although SR of BC is a rare event, it is important to know that it might happen. It is also of great importance to try to better explain, over time, its underlying mechanism. This knowledge could help us to further develop cancer prevention methods and predict the clinical course of these kinds of neoplasms.

The perfection of the seed is not enough to secure the development of the plant; the soil in which it is sown must be capable of feeding it.

## Figures and Tables

**Figure 1 diagnostics-13-01224-f001:**
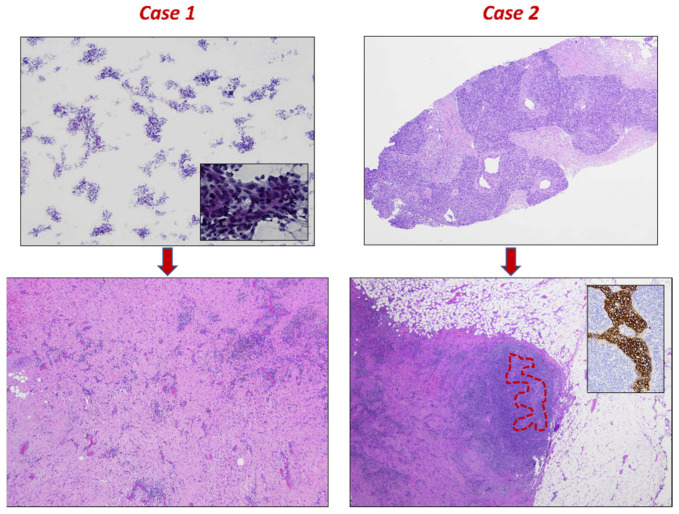
Fine-needle aspiration (**top left**, 2× magnification, Papanicolaou stain) and biopsy (**top right**, 2× magnification, hematoxylin & eosin stain) findings. Malignant epithelial tumors with high-grade nuclear features, which are more clearly visible in the inset (40× magnification, Papanicolaou stain). Post-operative histopathological samples show scar tissue with inflammation. There is no more of the tumor in the first case (**bottom left**, 4× magnification, hematoxylin & eosin stain), and just a small, residual, solid nest in the second case (**bottom right**, 2× magnification, hematoxylin & eosin stain), which is highlighted in red and by CKAE1/AE3 stain (inset, 20× magnification).

**Figure 2 diagnostics-13-01224-f002:**
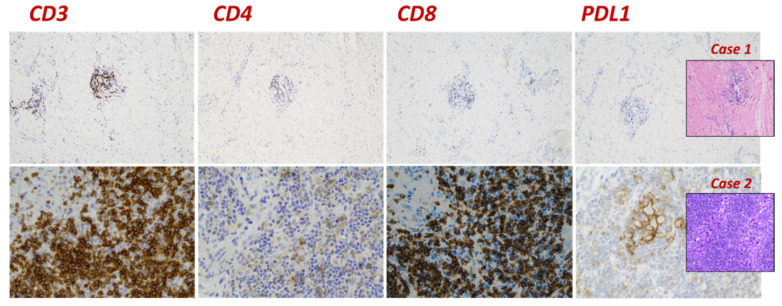
Immunohistochemistry results highlighting the inflammatory component in case 1 (**top row**) and in case 2 (**bottom row**). The greater part of inflammatory cells is represented by CD3-positive T lymphocytes, although there are both T helper CD4 (faint staining) and T cytotoxic CD8 (strong staining) lymphocytes; however, only the second case shows PD-L1 expression (22C3 DAKO–CPS > 10) on both inflammatory and neoplastic cells.

**Table 1 diagnostics-13-01224-t001:** Clinicopathological features of breast cancer–spontaneous regression.

Author	Year of Publication	Gender, Age	Diagnosis	Size (mm)	Size, Post (mm)	Phlogosis
MacKay, C.G.	1907	F, 37	Scirrhous cancer		Disappearance	
Hodenpyl, E.	1910	F, 37	Carcinoma			
Lilienthal, H.	1913	F, 61	Scirrhous cancer			
Scott, J.B.	1935	F, 40	Carcinoma			
Boyd, W.	1966	15 cases				
Maiche, A.G.	1994		Stage IV breast cancer			
Tokunaga, E.	2014	F, 52	Invasive ductal carcinoma (ER+, PgR−)	9	2	Yes
Maillet, L.	2014	F, 34 F, 82	TNBC		Disappearance	Yes
Ito, E.	2016	F, 44	Invasive ductal carcinoma (ER+, PgR−)	15	Disappearance	Yes
Nijjar, Y.	2018	M, 77	Merkel cell carcinoma	24	Disappearance	Yes, low
Takayama, S.	2019	F, 67	Occult metastatic BC, ER+, PgR+	3.5	Disappearance	Yes
Cserni, G.	2019	F, 72	TNBC with medullary-like features	16	Disappearance of invasive tumor with residual DCIS	Yes, high
Katano, K.	2020	M, 70	Invasive ductal carcinoma	12	Temporary regression	Yes, mild and focal
Ohara, M.	2021	F, 59	TNBC	30	3	Yes, post

## Data Availability

Additional data are available from the corresponding author upon reasonable request.

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
