# Peer review of "What Can Trigger Spontaneous Regression of Breast Cancer?"

_diagnostics, 2023, doi:10.3390/diagnostics13071224_

Round 1

Reviewer 1 Report

D'Alessandris et al. reported two cases of spontaneous regression of breast cancer. They also did a thorough literature search and found only 29 reported cases of breast cancer spontaneous regression thus far. They also discussed the proposed mechanisms of spontaneous regression including fever, acute infection, incomplete surgical removal, immunity, etc. Of all the mechanisms, cancer cells induced spontaneous T cell mediated response is the most widely accepted one. Although the exact mechanism of spontaneous breast cancer regression is still unclear, reporting more cases will contribute to the study and understanding of the actual mechanisms and aid the prevention, prognosis, and treatment of breast cancer. This paper is a well-written report and is ready to be published.

Author Response

We thank the reviewer for its positive comments.

Reviewer 2 Report

Comments and Suggestions for Authors:

Title: WHAT CAN TRIGGER THE SPONTANEOUS REGRESSION OF BREAST CANCER?

By analyzing two biopsy samples, the author reveals some reasons that can induce the spontaneous regression of breast cancer. It is an important study, but there are some problems, which must be solved before it is considered for publication.

1-      The induction does not completely show the current results of the study, it needs to rewrite.

2-      The results were not clearly presented, and these results do not support the conclusion.

3-      Authors need to check the spellings in current manuscript.

Author Response

We thank the reviewer for its comments and suggestions.

Herein is a point-by point reply to all comments. Please note that all changes in the manuscript have been highlighted in red.

Q1-      The induction does not completely show the current results of the study, it needs to rewrite.

A1: introduction has been modified and corrected as suggested.

Q2-      The results were not clearly presented, and these results do not support the conclusion.

A2: Results has been re-written to better explain all relevant findings.

Q3.   Authors need to check the spellings in current manuscript.

A3: We have checked and corrected all mistakes.

Reviewer 3 Report

Dear authors

The manuscript needs to be improved significantly, there are flaws to be overcome, regarding the research design.

Regarding the clinical cases, from lines 51 to 61, needs support, images and numbers, more than just sharing what was observed

There is a need to show images and quantification.

Table 1 is a supposition please remove the column possible trigger, no adding value to manuscript.

references are rather old, and your cases should not be included in table.

we do not have the inclusion and exclusion criteria for studies included in table.

Discussion must be deepened significantly

There is a need for a conclusion chapter.

All et al. must be in italics, once it is Latin.

Author Response

We thank the reviewer for its comments and suggestions.

Herein is a point-by point reply to all comments. Please note that all changes in the manuscript have been highlighted in red.

Q1: Regarding the clinical cases, from lines 51 to 61, needs support, images and numbers, more than just sharing what was observed.

A1: Results have been re-written to better explain all relevant findings. A new image (Figure 2) has also been included to better characterize the immune infiltrate of the reported cases.

Q2: There is a need to show images and quantification.

A2:  As suggested we modified and corrected Figure 1 and 2.

Q3: Table 1 is a supposition please remove the column possible trigger, no adding value to manuscript.

A3: we have corrected.

Q4: references are rather old, and your cases should not be included in table.

A4: As suggested, more recent references have been included in the manuscript; our cases have also been deleted in the table.

Q5: we do not have the inclusion and exclusion criteria for studies included in table.

A5: As suggested, inclusion/exclusion criteria have been included in methods section.

Q6: Discussion must be deepened significantly; There is a need for a conclusion chapter.

A6: Discussion has been re-written also including novel chapters. Conclusion chapter has also been included.

Q7: All et al. must be in italics, once it is Latin.

A7: we have corrected.

Round 2

Reviewer 2 Report

This revision looks good. authors can further check the spellings in proof stage.